# B7H3 Role in Reshaping Immunosuppressive Landscape in MSI and MSS Colorectal Cancer Tumours

**DOI:** 10.3390/cancers15123136

**Published:** 2023-06-10

**Authors:** Sylwia Mielcarska, Miriam Dawidowicz, Agnieszka Kula, Paweł Kiczmer, Hanna Skiba, Małgorzata Krygier, Magdalena Chrabańska, Jerzy Piecuch, Monika Szrot, Błażej Ochman, Julia Robotycka, Bogumiła Strzałkowska, Zenon Czuba, Dariusz Waniczek, Elżbieta Świętochowska

**Affiliations:** 1Department of Medical and Molecular Biology, Faculty of Medical Sciences in Zabrze, Medical University of Silesia, 19 Jordana, 41-808 Zabrze, Poland; s74817@365.sum.edu.pl (B.O.); s78516@365.sum.edu.pl (J.R.); s74852@365.sum.edu.pl (B.S.); eswietochowska@sum.edu.pl (E.Ś.); 2Department of Oncological Surgery, Faculty of Medical Sciences in Zabrze, Medical University of Silesia, 41-808 Katowice, Poland; d201069@365.sum.edu.pl (M.D.); d201070@365.sum.edu.pl (A.K.); dwaniczek@sum.edu.pl (D.W.); 3Department and Chair of Pathomorphology, Faculty of Medical Sciences in Zabrze, Medical University of Silesia, 13-15 3 Maja, 41-800 Zabrze, Poland; pkiczmer@sum.edu.pl (P.K.); hanna.skiba@sum.edu.pl (H.S.); mkrygier@sum.edu.pl (M.K.); mchrabanska@sum.edu.pl (M.C.); 4Department of General and Bariatric Surgery and Emergency Medicine in Zabrze, Faculty of Medical Sciences in Zabrze, Medical University of Silesia, 10 Marii Curie-Skłodowskiej, 41-800 Zabrze, Poland; jpiecuch@sum.edu.pl (J.P.); mszrot@sum.edu.pl (M.S.); 5Department of Microbiology and Immunology, Faculty of Medical Sciences in Zabrze, Medical University of Silesia, 19 Jordana, 41-808 Zabrze, Poland; zczuba@sum.edu.pl

**Keywords:** B7H3, colorectal cancer, immune tumour microenvironment (iTME)

## Abstract

**Simple Summary:**

Colorectal cancer (CRC) is one of the most prevalent malignant neoplasms worldwide, responsible for over 900,000 deaths yearly. As the immunotherapies targeting the PD-1/PD-L1 axis in CRC are effective only in microsatellite unstable tumours, which are 15% of all CRC cases, new targets of immune evasion are still needed. B7H3 has been reported to mediate immune escape and promote tumour progression in numerous malignancies, but it has yet to be fully elucidated in CRC. The study investigates whether B7H3 expression is related to MSI/MSS status, tumour infiltrating lymphocytes and cytokine composition in CRC. We found that B7H3 expression is upregulated in CRC tumours and independent of MSI/MSS status. B7H3 correlated positively with cytokines supporting tumour growth and was associated with M2-macrophage polarization. Additionally, TCGA analysis showed that high B7H3 expression in CRC tumours is related to decreased survival in CRC patients. Our findings provide a novel insight into B7H3’s role in CRC immunity.

**Abstract:**

The study aimed to assess the expression of B7H3 concerning clinicopathological and histological parameters, including MSI/MSS status, CD-8 cells, tumour-infiltrating lymphocytes (TILs), budding, TNM scale and grading. Moreover, we analyzed the B7H3-related pathways using available online datasets and the immunological context of B7H3 expression, through the 48-cytokine screening panel of cancer tissues homogenates, immunogenic features and immune composition. The study included 158 patients diagnosed with CRC. To assess B7H3 levels, we performed an immunohistochemistry method (IHC) and enzyme-linked immunosorbent assay (ELISA). To elucidate the immune composition of colorectal cancer, we performed the Bio-Plex Pro Human 48-cytokine panel. To study biological characteristics of B7H3, we used online databases. Expression of B7H3 was upregulated in CRC tumour tissues in comparison to adjacent noncancerous margin tissues. The concentrations of B7H3 in tumours were positively associated with T parameter of patients and negatively with tumour-infiltrating lymphocytes score. Additionally, Principal Component Analysis showed that B7H3 expression in tumours correlated positively with cytokines associated with M2-macrophages and protumour growth factors. The expression of B7H3 in tumours was independent of MSI/MSS status. These findings will improve our understanding of B7H3 role in colorectal cancer immunity. Our study suggests that B7-H3 is a promising potential target for cancer therapy. Further studies must clarify the mechanisms of B7H3 overexpression and its therapeutic importance in colorectal cancer.

## 1. Introduction

Immunotherapy is a recently discovered treatment that stimulates or inhibits the immune system from intensifying or decreasing an immune response and has rapidly become a significant treatment for multiple types of solid cancers [1].

In colorectal cancer, immune checkpoint therapy is approved for treating tumours with a mismatch repair deficiency (MMR-D) and microsatellite instability-high (MSI-H) phenotype [2]. MSI results from mutations in the DNA mismatch repair genes and is found in 15% of sporadic colorectal cancers [3].

Present immune checkpoint inhibitors (ICIs) are ineffective in tumours that are microsatellite-stable (MSS) or have low levels of microsatellite instability (MSI-L) [4]. Research on new inhibitory receptors for immunotherapy is essential to expand the patient group that benefits from immunotherapy.

The B7 family molecules have received growing attention in recent years and are essential for regulating T-cell responses. The members of the B7 family function as immune regulatory ligands that exhibit interaction with the CD28 receptors family [5,6]. B7H3 (CD276) is a newly found immune checkpoint of the B7 family, representing a good target for cancer immunotherapy. The human B7-H3 is a type 1 transmembrane glycoprotein with two isoforms: 2IgB7-H3 (B7-H3 VC) and 4IgB7-H3 (B7-H3 VCVC) [7,8]. The 2IgB7-H3 structure incorporates single extracellular V- and C-like Ig domains, a transmembrane region, and a 45-aa cytoplasmic tail. The 4IgB7-H3 isoform contains two identical pairs of IgV-like and IgC-like domains (four Ig-like domains), so this isoform is mostly expressed on the cell surface in human mononuclear cells and cancer cells [9,10]. The FG loop of the IgV domain of B7H3 [11] is critical for inhibiting T-cell proliferation. B7H3, as an immune checkpoint, is overexpressed on malignant cells with limited heterogeneity [12]. It was reported that B7H3 has a dual role in regulating the immune system, playing both costimulatory and coinhibitory functions. As a costimulator of the immune system, B7H3 promotes cellular immunity and upregulates IFNγ production in the presence of TCR signalling. The inhibitory functions of the studied immune checkpoint are primarily associated with suppressing T-cell activation and proliferation through NFAT, NFκB and AP-1 factors, the primary regulators of TCR gene transcription [13]. The knowledge about the role of B7H3 in colorectal cancer is insufficient and needs more investigation. Importantly, previous studies demonstrated that B7H3 has a crucial role in promoting epithelial-to-mesenchymal transition (EMT), invasion, metastasis and chemotherapy resistance in CRC [14,15,16]. 

Here, we aimed to assess the expression of B7H3 regarding clinicopathological parameters, including MSI/MSS status, CD8+ T-cells, histopathological features: budding, tumour-infiltrating lymphocytes (TILs), TNM scale, and grading. Moreover, we examined the B7H3-related pathways using available online datasets. Our research also aimed to investigate the immunological context of B7H3 expression through the 48-cytokine screening panel, immunogenic elements, and immune landscape.

## 2. Materials and Methods

### 2.1. Characteristics of the Patient Group

The samples used in the study were obtained from 158 patients who had undergone surgery due to CRC. Patients were operated in the 1st Specialistic Hospital in Bytom, Poland (approval of the Research Ethics Committee No. PCN/0022/KB1/42/VI/14/16/18/19/20, 14 July 2020). The obtained specimens included colorectal tumour tissues and adjacent “tumour-free” surgical margins, confirmed in histological examination. The qualification for the study was led according to the inclusion and exclusion criteria described in our previous studies [17,18]. The characteristics of the research group are presented in Table 1.

### 2.2. Evaluation of the B7H3 Concentrations with ELISA

To weigh and homogenize examined tumour and surgical margin tissue, we have followed our previous research papers’ homogenization protocol [18,19]. We conducted an enzyme-linked immunosorbent assay (ELISA) to determine the concentration of the B7H3 protein. The B7H3 levels were measured using a human B7H3 ELISA kit (Cloud Clone, Wuhan, China), as per the manufacturer’s manual, with a sensitivity of 0.118 ng/mL. The absorbance of the samples was measured at a wavelength of 450 nm using a Universal Microplate Spectrophotometer (μQUANT, Biotek Inc., Winooski, VT, USA). The results were recalculated to correspond to the total protein level and demonstrated as ng/mg of protein.

### 2.3. Evaluation of the B7H3 Expression by IHC

B7H3 expression was assessed in 77 randomly selected cases using B7H3 immunostaining. The tissue specimens from formalin-fixed paraffin-embedded tissue blocks were deparaffinized and rehydrated. Subsequently, antigen retrieval was conducted by incubating slides in EnVision Flex Target Retrieval Solution High pH (Dako, Carpinteria, CA, USA) at 95 °C for 20 min. The prepared samples were treated with Peroxidase-Blocked Reagent (Dako) and then exposed to the B7H3 Polyclonal Antibody (Invitrogen, Waltham, MA, USA) with a 40 min incubation time and a dilution of 1:300. EnVision FLEX HRP (Dako) was applied afterwards. Later, 3,3′-diaminobenzidine was used to stain the antigen-antibody complexes. Next, the tissue sections underwent counterstaining with hematoxylin, and dehydration and were covered with coverslips for subsequent analysis. Two independent pathologists conducted a histological evaluation using an Olympus BX51 microscope. Tumours were classified as B7H3 positive if staining was detected in 1% or more cancer cells. Furthermore, the percentage of tumour cells exhibiting positive B7H3 staining was utilized for additional statistical analyses.

### 2.4. Assessment of the MSI/MSS Status

To determine the MSI/MSS status, immunohistochemical (IHC) staining for MSH2, MSH6, PMS2, and MLH1 was conducted following the procedure described in our previous publications [20,21]. A representative formalin-fixed, paraffin-embedded (FFPE) tumour tissue block, cut into 4-µm thick sections, was utilized on a Dako Autostainer Link 48. The samples underwent deparaffinization and rehydration, followed by antigen retrieval through incubation of the slides in EnVision Flex Target Retrieval Solution High pH (Dako, Carpinteria, CA, USA) at 95 °C for 20 min. Furthermore, the specimens were processed with Peroxidase-Blocked Reagent (Dako), followed by exposure to specific antibodies: Mouse Monoclonal antibody MSH2 (G219-1129), Cell Marque, with a 30 min incubation time and a dilution of 1:400; Mouse Monoclonal antibody MSH6 (44), Cell Marque, with a 45 min incubation time and a dilution of 1:100; Mouse Monoclonal antibody PMS2 (MRQ-28), Cell Marque, with a 40 min incubation time and a dilution of 1:50; Mouse Monoclonal antibody MLH1 (G168-728), Cell Marque, with a 40 min incubation time and a dilution of 1:100. EnVision FLEX HRP (Dako) was applied subsequently. The 3,3′-diaminobenzidine was used to stain the antigen-antibody complexes. Finally, the collected tissue specimens underwent hematoxylin counterstaining, dehydration, and were subsequently covered with coverslips for subsequent evaluation.

The nuclear staining of invasive tumour cells for MSH2, MSH6, PMS2, and MLH1 was evaluated in an internal positive control (inflammatory and stromal cells) to assess the tumours. The tumours showing nuclear staining for at least 1% of invasive tumour cells were classified as having positive marker staining. The interpretation of immuno-histochemistry testing followed the criteria established by Olave and Graham [22] and described in our previous articles [20,23]. The MSI status was confirmed if any of the following marker patterns were observed: loss of MLH1 and PMS2, loss of PMS2 alone, loss of MSH2 and MSH6, or loss of MSH6 alone.

### 2.5. Evaluation of the Tumour-Infiltrating CD8+ T-cells

For immunohistochemical (IHC) analysis, tissue sections measuring 4 µm in thickness were utilized. These sections underwent deparaffinization using xylene, rehydration in a series of graded alcohol, and thorough rinsing with deionized water. Subsequently, antigen retrieval was achieved by incubating the slides in EnVision Flex Target Retrieval Solution High pH (Dako, Carpinteria, CA, USA) at 95 °C for 20 min. Prepared samples were then incubated with a peroxidase-blocked reagent (Dako), followed by treatment with the CD8+/144B mouse monoclonal antibody diluent. The incubation time was set for 40 min, with a dilution ratio 1:100. Subsequently, the samples were exposed to EnVision FLEX HRP (Dako). The antigen-antibody complexes were then subjected to staining with 3,3′-diaminobenzidine, which led to their visualization. Lastly, the tissue sections were counterstained with hematoxylin, dehydrated, and covered with coverslips for subsequent analysis, as described in the previous publications [20,23].

### 2.6. Assessment of the TILs and Budding

The extent of lymphatic infiltration associated with the tumour was assessed in a semi-quantitative manner on the same H&E-stained slides using a five-grade scale, as per the guidelines established by Salgado et al. in breast cancer [24], which were also applied in our previous studies [20,23]. This evaluation considered various aspects of lymphocyte presence, including intratumoural lymphocytes with direct contact between lymphocytes and tumour cells, as well as stromal tumour-infiltrating lymphocytes (TILs) dispersed within the tumour tissue without direct contact, including TILs in the invasive margin. Stromal TILs were scored following the recommended methodology based on the percentage of the stromal area alone, excluding areas occupied by carcinoma cells. Lymphatic infiltrates outside the tumour boundaries were not included in the assessment. TILs were categorized as TILs 1 when the lymphocyte infiltration area was less than 5%, while TILs 2, TILs 3, and TILs 4 corresponded to 5–25%, 25–50%, and 50–75% of lymphocytes in the stroma, respectively. TILs 5 was defined as more than 75% lymphocyte infiltration. Tumour budding analysis was also conducted in the same cases, focusing on the invasive front and assessing the number of buds in FOV under ×20 magnification. The bud count was adjusted using the normalization factor (1.210). Based on the budding assessment, tumours were categorized as having low budding (0–4 buds), intermediate budding (5–9 buds), or high budding (>10 buds). The mean number of buds per FOV was also used in further statistical analysis.

### 2.7. Evaluation of the Cytokines Screening Panel

The levels of chemokines/cytokines/growth factors were assessed in 77 tissue homogenates using the Bio-Plex Pro Human cytokines screening panel 48 cytokines assay (Bio-Rad Laboratories, Hercules, CA, USA) following the manufacturer’s procedure. Briefly, 50 µL aliquot of tissue homogenate was diluted 1:2 with sample diluent, incubated with anti-body-coupled beads, biotinylated secondary antibodies, and subsequently with streptavidinphycoerythrin. Standard curves for each examined molecule were performed using corresponding cytokine standard solutions. The beads were measured using the Bio-Plex 200 System. The intra-assay %CV varied up to 15%, and the inter-assay %CV varied up to 25%, depending on the studied protein. We transformed the obtained levels to the corresponding total protein concentrations. This method was applied in our previous research [20,23].

### 2.8. Exploration of B7H3 Biological Functions

Functional annotation analysis based on mRNA expression profiles was conducted on the CRC online dataset obtained from the FieldEffectCrc Package, specifically among cohort A consisting of 311 CRC samples [25]. The DESeq2 package [26] was employed to normalize the matrix data. The cohort was stratified into groups based on high and low expression B7H3 levels. Gene set enrichment analysis was employed to explore potential hallmarks pathways associated with B7H3 in CRC, utilizing the Molecular Signatures Database (h.all.v7.5.symbols.gmt) within R Studio and the fgsea package. Genes exhibiting significant differences between high and low B7H3 expression groups were selected for GO enrichment analyses (|logFC| > 0.5 and *p*.adj. < 0.05). For elucidating the association between B7H3 expression, immune cell infiltration, Mantis score, and patient survival, the web-based tool CAMOIP was utilized to analyze TCGA-COAD data (http://220.189.241.246:13838/#shiny-tab-188home (accessed on 1 March 2023)).

### 2.9. Statistical Analysis

The distribution of the analyzed data were evaluated utilizing the Shapiro–Wilk test. For a better fit to the Gaussian distribution, a logarithmic transformation was applied to the levels of the analyzed molecules. Mean ± SD was used to present variables with a normal distribution, while median with interquartile range was used for variables with a non-normal distribution. To compare tumour and margin levels, the paired Student’s *t*-test (for variables with a normal distribution) and paired samples Wilcoxon test (for variables with a non-normal distribution) were applied. Independent variables were compared using the Student’s *t*-test and Mann–Whitney U test. Correlations between the examined variables were assessed using Pearson’s or Spearman’s coefficients, depending on the distribution of the variables. The relationship between the levels of the studied proteins and the TNM scale was determined using Tau-Kendall’s tau rank correlation coefficient. Principal component analysis was carried out to decrease the number of correlated variables and elucidate the connections between B7H3 expression and cytokine concentrations obtained from the used screening panel. The principal components exhibiting the greatest explained variance were employed as fresh variables for subsequent analyses. Significance at a statistical level was defined as *p*-values ≤ 0.05. The statistical analysis was executed using STATISTICA 13 software (Statsoft, Tulsa, OK, USA) and R Studio (Integrated Development for R. RStudio, PBC, Boston, MA, USA).

## 3. Results

### 3.1. The B7H3 Expression Is Upregulated in Tumour Tissues

In this research, to investigate the expression of B7H3 in CRC patients, the B7H3 concentrations were determined in tumour tissue and normal colon mucosa adjacent to CRC. In the 154 pairs of tumour tissues and adjacent tumour-free surgical margins, the B7H3 levels were assessed with an ELISA test. The concentrations of B7H3 were significantly higher in the tumour tissue compared to the margin tissue (Figure 1). 

Furthermore, the B7H3 levels in tumour tissue homogenates correlated positively with the T parameter and were negatively associated with the TILs score (Figure 2).

Additionally, we investigated the expression of B7H3 using IHC staining in randomly selected 77 tumour section slides. 29.77% of all cases were positive for B7H3 staining (Table 2, Figure 3). The B7H3 immunohistochemical expression in tumour cells was independent of MSI/MSS status (Chi-square test, *p* = 0.52). We did not observe any association between B7H3 IHC expression in tumour cells and clinicopathological features of the patients (Table 2).

### 3.2. B7H3 Expression and Cytokines Landscape in CRC Tumours

In our study, we used Bio-Plex Pro Human Cytokine screening panel containing cytokines, chemokines and growth factors to elucidate the immunological context of B7H3 expression. Correlations between cytokines and B7H3 tumour concentrations are presented in Figure 4 as a heatmap with a dendrogram (Figure 4).

The cytokine screening panel was divided into a group of cytokines associated with M2 macrophages (IL-1b, IL-1ra, IL-1a, TNF-alpha, IL-10, IL-6, IL-8, VEGF, FGF-basic) (Figure 5, Figure 6 and Figure 7, Table 3 and Table 4) and a group containing protumour cytokines (IL-1β, IL-1Rα, IL-4, IL-5, IL-8, IL-9, IL-10, IL-13, IL-17A, TNFβ) (Figure 8 and Figure 9, Table 5 and Table 6) to perform principal component analysis (PCA). Factor 1 from PCA for cytokines with M2 macrophages was positively associated both with B7H3 concentrations in tumour tissue homogenates and with the percentage of B7H3 IHC expression in tumours (Figure 7). Additionally, that factor also correlated positively with the T parameter of patients (Figure 7). 

In PCA for protumour cytokines (Figure 8 and Figure 9, Table 5 and Table 6), factor 1 was positively associated with concentrations of B7H3 in tumour tissue homogenates (Figure 10).

### 3.3. Exploration of Survival Data and B7H3 Immune Infiltration Landscape Based on TCGA-COAD Data

CAMOIP allows us to perform survival analysis according to the low or high expression level of each gene in the TCGA cohort. To investigate the difference in overall survival between individuals with high and low B7H3 expression in the TCGA COAD cohort, we performed a survival analysis in CAMOIP. The patients with high B7H3 expression showed significantly decreased overall survival time compared to the low expression group (Figure 11).

To elucidate the link between B7H3 expression and immune infiltration landscape in CRC tumours, we used CAMOIP- a tool for analyzing the expression and mutation data from TCGA which provides comprehensive analysis on potential immunotherapy targets. Immune Infiltration cells analysis was estimated using CIBERSORT algorithm. The percentages of CD4+ memory resting T-cells, CD4+ memory activated T-cells were lower in CRC tumours with upregulated B7H3 expression, while conversely an increased fraction of regulatory T-cells (Tregs), monocytes, macrophages M0, eosinophils and neutrophils was associated with higher B7H3 expression (Figure 12). The immune scores of Th1 cells, Stromal fraction, macrophage regulation, Lymphocyte Infiltration Signature Score, IFN-gamma Response, TGF-beta response were also increased in tumours with B7H3 high expression, while Th2 Cells and Aneuploidy score were down-regulated (Figure 12). Additionally, the expression of CD163, a M2 macrophages marker, was elevated in tumours with increased B7H3 expression (Figure 12).

### 3.4. Analysis of Functional Annotations and B7H3-Induced Signaling Pathways in CRC

To determine pathways activated in tumours with upregulated B7H3 expression, gene set enrichment analysis (GSEA) was performed to analyze the difference in gene expression between low and high B7H3 expression datasets. Significantly upregulated and downregulated pathways (*p*. adjusted < 0.05) in the enrichment of the Molecular Signature Database Collection for Hallmark gene sets are provided in Figure 13 and Figure 14. The results indicated that the most upregulated pathways for high B7H3 expression included oxidative phosphorylation, fatty acid metabolism, adipogenesis, bile acid metabolism, heme metabolism, pancreas beta cells, and genes downregulated by KRAS activation. These findings could suggest that tumours with high B7H3 expression may exhibit impaired intracellular processes and change in KRAS signalling. On the contrary, elevated expression of B7H3 was related to downregulation in pathways associated with hypoxia, glycolysis, interferon alpha response, MTORC1 signalling, G2M checkpoint and E2F targets suggesting a potential link between B7H3 and altered cell cycle regulation (Figure 13).

## 4. Discussion

Immune Checkpoint Therapy (ICT) is a new approach to cancer treatment aimed at strengthening anti-tumour response mediated by T-cells and is characterized by high efficiency. B7 family and its most known member—PD-L1—are known for regulating immune response mediated by T-cells and exhibiting significant antitumour effect, which has been widely described in several malignancies. Due to the limited possibilities of therapies targeting the PD-1/PD-L1 axis in CRC, new members of the B7 family, including B7H3, are widely investigated. B7H3, a recently discovered protein belonging to the B7 family that acts as a T-cell inhibitor promoting tumour cell invasiveness rather than a T-cell stimulator as previously classified, has gained increasing attention in the anti-cancer battlefield [27]. Numerous receptors have been reported to interact with different B7 family members, but the B7H3 receptor remains unexplored. Previously, there was some evidence that Trem-like transcript 2 (TREML2) could act as a putative B7H3 receptor, but the results were controversial [28,29]. Currently, the Interleukin receptor IL20RA is considered a possible B7H3 receptor, however, confirmation research is still ongoing [30]. It seems that controversies about the costimulatory or coinhibitory role of B7H3 may be associated with an unidentified B7H3 receptor. 

On the mRNA level, B7H3 is widely expressed in most normal tissues, but the presence of B7H3 protein is strictly limited, suggesting a significant role in post-transcriptional regulation of the B7H3 gene [31]. On the contrary, B7H3 expression in tumour tissue is highly upregulated, which was also confirmed in CRC studies [32]. Similarly to these previous findings, we also found significantly higher levels of B7H3 protein in CRC tumour tissue compared with normal adjacent mucosa, in contrast to the percentage of tumours that displayed positive staining for B7H3 in immunohistochemistry, which was 30% for all study cohorts. The percentage of B7H3-positive tumours differs considerably between particular studies. Sun et al. and Mao et al. reported 50% positive B7H3 tumours in their cohort, while according to other authors, B7H3 expression ranged to more than 80% CRC tumours [32,33]. Furthermore, we observed that B7H3 expression was independent of MSI/MSS status. These findings are consistent with the data obtained from CAMOIP, showing that differences in MANTIS score predicting MSS/MSI status of tumour are much lower for B7H3 compared with PD-L1, for which a close link with MSI status was well confirmed in CRC. Additionally, Zhao et al. did not report an association between B7H3 expression and MS/MSS status in the CRC cohort [16]. 

We explored the relationship between B7H3 expression and the clinicopathological parameters of the patients. The results showed a positive correlation between B7H3 tumour concentration and the T parameter, thus suggesting that B7H3 contributed to CRC progression. Similarly, Ingebrigtsen et al. reported that high B7H3 expression was associated with the advanced TNM stage in CRC [15]. Furthermore, a positive relationship between B7H3 expression and TNM stage was confirmed in a variety of solid tumours, including breast, lung, and kidney tumours [27]. In several studies, B7H3 was proven to play a crucial role not only in evasion from immune surveillance but also in non-immune processes promoting malignancy progression, such as epithelial-mesenchymal transition, chemoresistance, cell survival, migration and invasion, which makes B7H3 an even more appealing therapeutic target for CRC [27,34]. 

As we suspected, B7H3 levels in the tumour were negatively correlated with tumour infiltrating lymphocytes (TILs) score. In CRC, tumour-infiltrating lymphocytes (TILs) consist of different populations of cells (T lymphocytes, B lymphocytes, NK cells) and are regarded as positive prognosis factors [35]. B7H3 was negatively related to TILS score in lung cancer [36]. At the same time, other members of the B7 family, such as PD-L1 and B7H4, were reported to correlate negatively with TILs in colorectal and gastric cancer [20,37]. While B7H3 tumour concentration was negatively associated with TILs score, there was no association between B7H3 expression and the density of CD8+ T-cells infiltrating a tumour. CD8 T-cells are the main effector of antitumour response partially controlled by immune checkpoints, which affect the activation of antigen-specific T-cells [38]. During malignant development, coinhibitory immune checkpoints allow tumour cells to escape from immune surveillance by accelerating T-cells exhaustion, thus, immune checkpoint blockade was found to induce long-term antitumour response mediated by CD8+ T-cells [38]. In addition, high expression of B7H3 was shown to be associated with a lower number of CD8+ T-cells in osteosarcoma, endometrial, and lung cancers [36,39,40]. Still, other CRC studies did not confirm this link [16,41]. 

Tumour-associated macrophages (TAMs) that promote tumour invasion and metastasis elicit phenotypes of M2-polarized macrophages [41]. Macrophage polarization is a process in which macrophages obtain distinct functional phenotypes (M1 and M2) in response to different microenvironmental signals and factors [42]. Macrophages M1 induced by interferon-gamma and LPS can kill tumour cells and pathogens, whereas macrophages M2 induced by IL-4, IL-10, and IL-13 exhibit anti-inflammatory activity and contribute to tumour invasion and angiogenesis [41,43]. Immune checkpoints, including B7 family members in the tumour microenvironment, are suspected of promoting TAMs polarization into M2 phenotypes. Performing Principal Component Analysis among molecules associated with M2-macrophages polarization (IL-1b, IL-1ra, IL-1a, TNF-alpha, IL-10, IL-6, Il-8, VEGF, FGF-basic), we found a positive correlation between Factor 1 obtained from PCA and B7H3 expression both in tissue homogenates and IHC staining. Factor 1 was also positively associated with the T parameter of patients. Similarly, CAMOIP analysis revealed that the M2 macrophage marker, CD163, is significantly higher in tumours with high B7H3 expression. However, a significant difference in immune cell scores between B7H3 high and low-expression tumours was found for M0 macrophage populations, but not M2 macrophages (CIBERSORT analysis). Previous studies have found an association between B7H3 expression and M2 polarization of macrophages in CRC, hepatocellular carcinoma and ovarian cancer [41,44,45]. Furthermore, it has been demonstrated that M2 polarization induced by B7H3 is mediated by the JAK2-STAT3 pathway in multiple myeloma [46]. These findings suggest that M2 TAMs may play a crucial role in mediating the immunosuppressive effects of B7H3. 

Exploring the cytokine network in the CRC tumour microenvironment may augment the current understanding of interrelationships between cytokines, chemokines, growth factors, and immune checkpoints and their common influence on TME modulation and tumour progression. As immune checkpoints blockade may reshape TME and restore antitumour response, investigating associations between B7H3 expression and functional groups of TME molecules could be extremely valuable for predicting the effects of B7H3 targeted therapy. Analysis of cytokines showed that B7H3 is related to protumour cytokines including IL-1β, IL-1Rα, IL-4, Il-5, IL-8, IL-9, IL-10, IL-13, IL-17A and TNFβ. Factor 1 obtained from PCA containing these molecules was positively related to B7H3 expression. IL-1β secreted by TAMs and neutrophils was found to promote cancer cell proliferation and recruit myeloid-derived suppressor cells (MDSCs) supporting tumour growth [47]. IL-4 expressed by Th2 cells contributed to cancer cell proliferation, epithelial-mesenchymal transition and metastasis [47]. Since IL-13 shares the same receptor with IL-4 (IL-4R alpha), its biological effects in CRC are similar to IL-4. IL-8 is involved in numerous processes supporting tumour progression, including proliferation, migration, invasion, survival of CRC cells and tumour angiogenesis [48]. IL-10, produced mainly by M2 macrophages and Th2 cells, shapes immunosuppressive TME by reducing the production of proinflammatory cytokines and Th1 antitumour cytokines, inhibiting the proliferation of T-cells and decreasing the expression of MHC class II antigens [48]. CD4+ Th17 cells are the main source of IL-17 which exhibits an immunosuppressive effect in TME by recruiting MDSCs, increasing CRC cell proliferation, upregulating VEGF production and activating oncogenic STAT-3 factor [48,49]. TNF-beta signalling was shown to be associated with tumour cell proliferation, epithelial-mesenchymal transition, invasion and metastasis formation [50]. Finally, the role of IL-5 and IL-9 remains controversial due to their anti- and protumour effects in several studies [51,52].

We also used CAMOIP, a web-based tool analyzing the TCGA-COAD dataset, to explore the immune composition of CRC according to high and low B7H3 expression. We found that upregulation of B7H3 affects only a few populations of immune cells, including CD4+ memory resting T-cells, CD4+ memory activated T-cells, regulatory T-cells, monocytes, macrophages M0, eosinophils and neutrophils. However, the immune score for Th1 cells, stromal fraction, macrophages regulation, lymphocytes infiltration, IFN gamma response and TGF beta was higher in a group with high B7H3 expression, indicating B7H3 contribution to immunological processes associated with tumourigenesis. Additionally, high B7H3 expression was associated with decreased survival in the TCGA-COAD dataset. This finding is consistent with clinical studies in which B7H3 was demonstrated to predict shorter overall survival and disease-free survival in CRC cohorts [38]. 

Analyzing the FieldEffectCrc dataset, we explored functions, processes and upregulated pathways associated with high B7H3 expression using GSEA and GO enrichment analysis.

Enrichment analysis showed that B7H3 is related to upregulated pathways involved in oxidative phosphorylation, fatty acid metabolism and downregulation of KRAS signalling. Recently, Picarda et al. demonstrated that B7H3 may play a pivotal role in regulating adipose tissue metabolism. B7H3 expression was upregulated in adipose tissue, with the highest levels in adipocyte progenitor cells and lower levels with the initiation of adipocyte differentiation. Additionally, the knockout of the B7H3 gene in progenitor cells led to impairment in aerobic metabolism and accumulation of fatty acids [53]. Mutation in the KRAS gene was found to be more frequent in B7H3-positive lung adenocarcinoma, however, data regarding CRC are missing, and the link between B7H3 and KRAS needs further research and explanation. 

Furthermore, high B7H3 expression is associated with downregulating pathways related to protooncogene MYC and cell-cycle control systems, such as G2/M Checkpoint and E2F target. Downregulation of the G2/M Checkpoint and E2F target indicates that one of the effects of B7H3 could be impaired G2/M phase cell cycle arrest. However, literature data about the association of B7H3 and the MYC gene are limited and concern only malignant gliomas in which a knockout of the B7H3 gene inhibits MYC expression [54]. 

Functional analyses of B7H3 revealed its involvement in immunological-associated processes. Significantly enriched GO terms were related to various immunological functions, including innate and adaptive immunity, such as phagocytosis, complement activation, antigen binding, activation of B-cells and their receptors, and regulation of immunoglobulin complex.

Our findings improved our understanding of B7H3′s role in cancer immunity. In addition, the studies on colorectal tumour tissues allowed us to estimate clinical, pathological, and immunological parameters. We assessed B7H3 concentrations by ELISA, and IHC, determined MSI/MSS status, and estimated the immune landscape by cytokines screening panel and tumour-infiltrating CD8+ T-cells. The analysis based on mRNA expression profiles has significantly expanded the scope of research by reporting many significant molecular roles of B7H3. CAMOIP, a tool for comprehensive analysis of potential immunotherapy targets, allowed us to perform survival analysis according to low or high B7H3 expression in the TCGA cohort and determined the link between B7H3 expression, immune cell infiltration, Mantis score, and patients’ survival in CRC tumours. We studied the pathways activated in tumours with upregulated B7H3 expression with gene set enrichment analysis. Our future research direction focuses on further studies of B7H3, as a potential target for immunotherapy which requires extending research to cellular, animal, and clinical models. The other cancer therapy trends require studying the diagnostic and therapeutic potential of B7-H3 thoroughly, its cellular and molecular mechanisms, B7H3-mediated metastases, tumour-associated vasculature, and recurrence. Clinical data with a long-term follow-up are essential to understand the relevance of B7H3 expression.

## 5. Study Limitations

There are some potential limitations in this research. The first one concerns using cytokines screening panels and in silico analyses to explore the immune landscape of CRC tumours without indirect assessment of selected immune cell populations through immunostaining or flow cytometry. Furthermore, the study has an observational character, and the main findings should be further evaluated using cell lines and animal tumour models.

## 6. Conclusions

Our study showed that B7H3 is overexpressed in CRC tumours and is independent of MSI/MSS status. High concentrations of B7H3 were positively associated with the patients’ T parameter suggesting its role in promoting tumour growth. Additionally, B7H3 may support tumour progression by shaping immunosuppressive tumour microenvironment mediated by M2-like macrophages and anti-inflammatory cytokines. The findings of this study suggest that B7-H3 is a promising potential target both in MSI and MSS CRC tumours. 

## Figures and Tables

**Figure 1 cancers-15-03136-f001:**
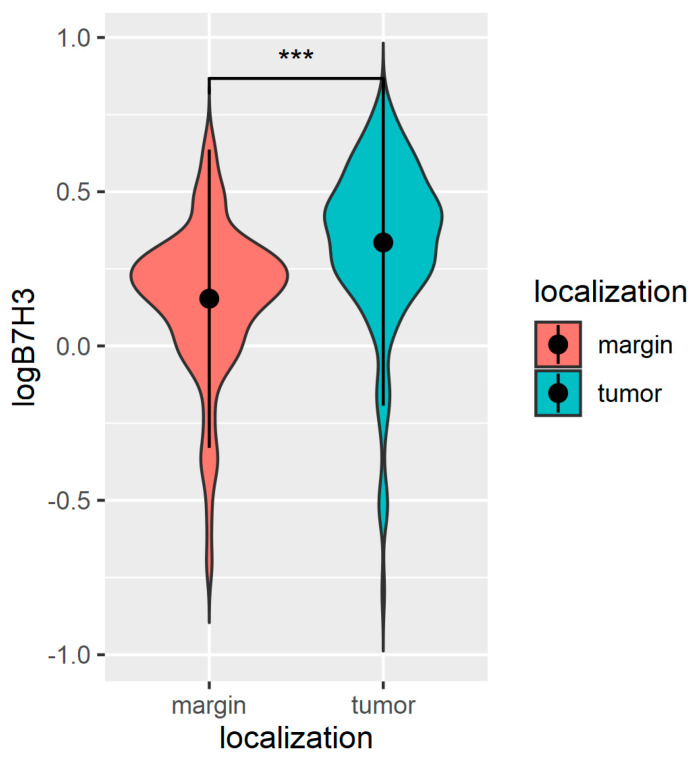
Violin plot of B7H3 levels in tumour and margin tissue (ng/mg). Wilcoxon signed-rank test *** *p* < 0.001.

**Figure 2 cancers-15-03136-f002:**
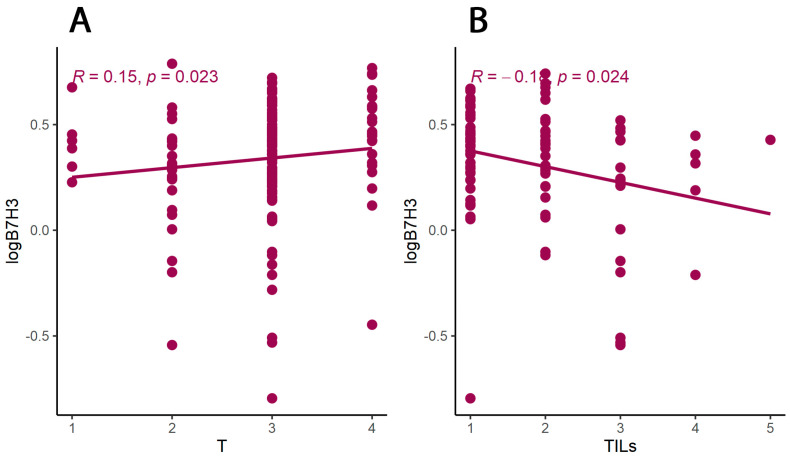
(**A**) Correlations of the B7H3 tumour concentrations and T parameter of patients. (**B**) Correlation of the B7H3 tumour concentration and TILs score. Tau Kendall correlation coefficient.

**Figure 3 cancers-15-03136-f003:**
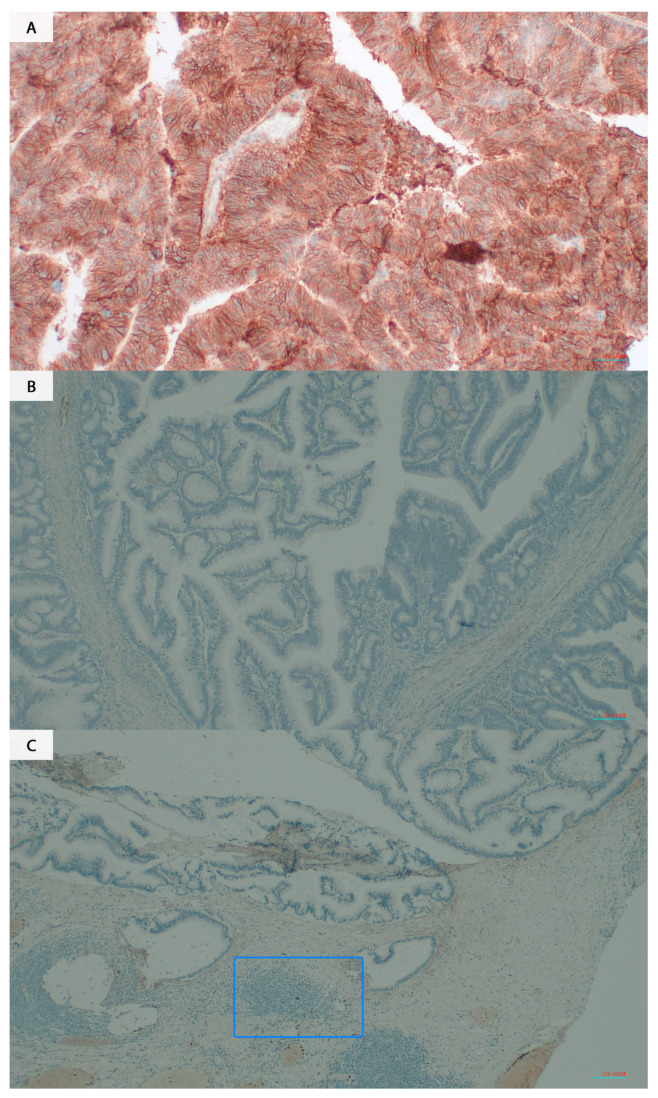
B7H3 immunostaining in CRC specimens. (**A**) B7H3 positive staining (**B**,**C**) B7H3 negative staining. (**C**) Tumour infiltrating lymphocytes are marked in frame (Opta Tech 2200 Camera (Opta Tech, Warsaw, Poland), magnification 200×).

**Figure 4 cancers-15-03136-f004:**
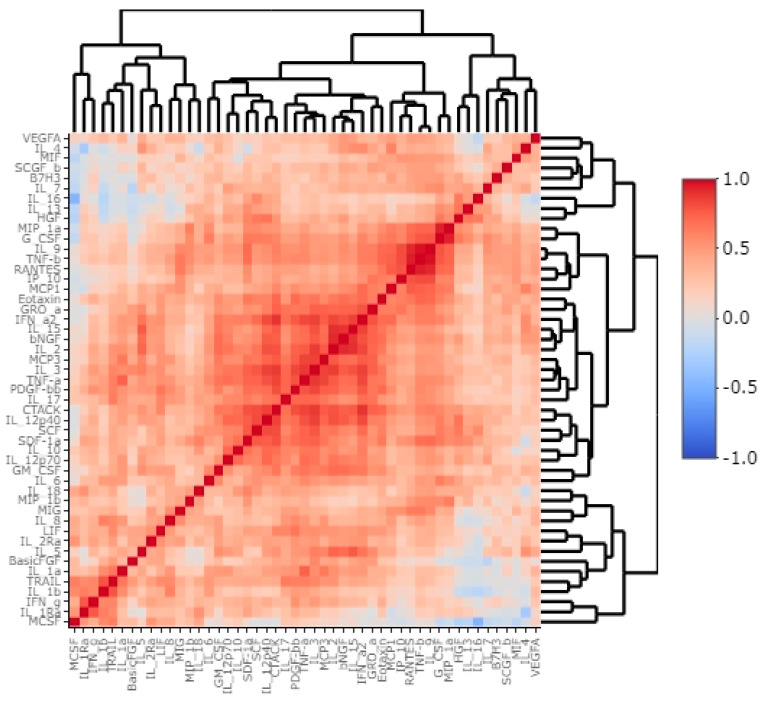
Correlation heatmap with dendrogram of molecules levels from cytokine screening panel and concentrations of B7H3 in tumour tissue homogenates.

**Figure 5 cancers-15-03136-f005:**
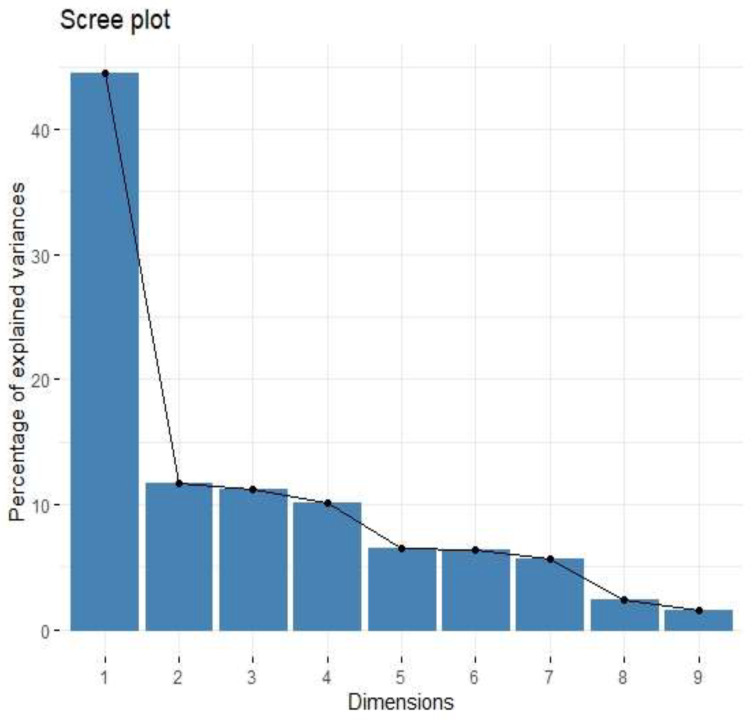
Scree plot of percentage of explained variances for nine principal components from the PCA for cytokines associated with M2 macrophages.

**Figure 6 cancers-15-03136-f006:**
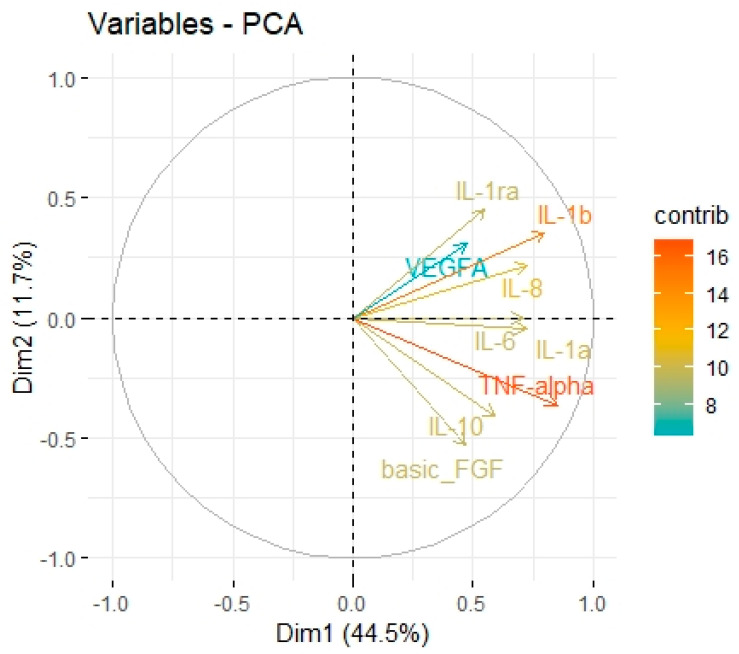
Principal Components 1 and 2 representing variance in two dimensions from the PCA for cytokines associated with M2 macrophages.

**Figure 7 cancers-15-03136-f007:**
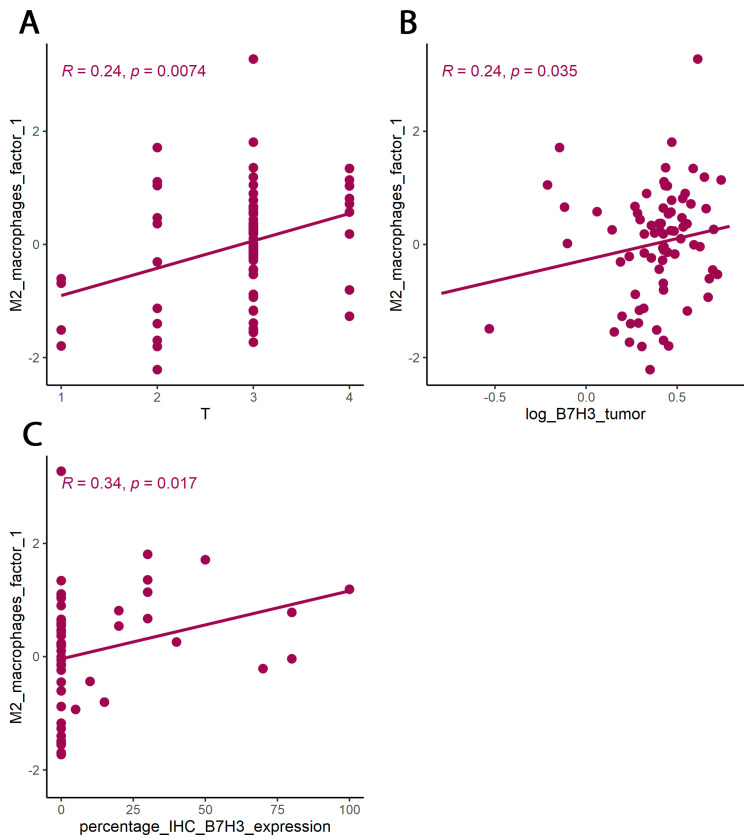
PCA for cytokines associated with M2 macrophages. (**A**) Correlation between M2 macrophage factor 1 (principal component 1) and T parameter (Kendall rank correlation coefficient). (**B**) Correlations between M2 macrophage factor 1 (principal component 1) and concentration of B7H3 in tumour tissue homogenates. (**C**) correlation between M2 macrophage factor 1 (principal component 1) and percentage IHC B7H3 expression in tumours (Spearman correlation coefficient).

**Figure 8 cancers-15-03136-f008:**
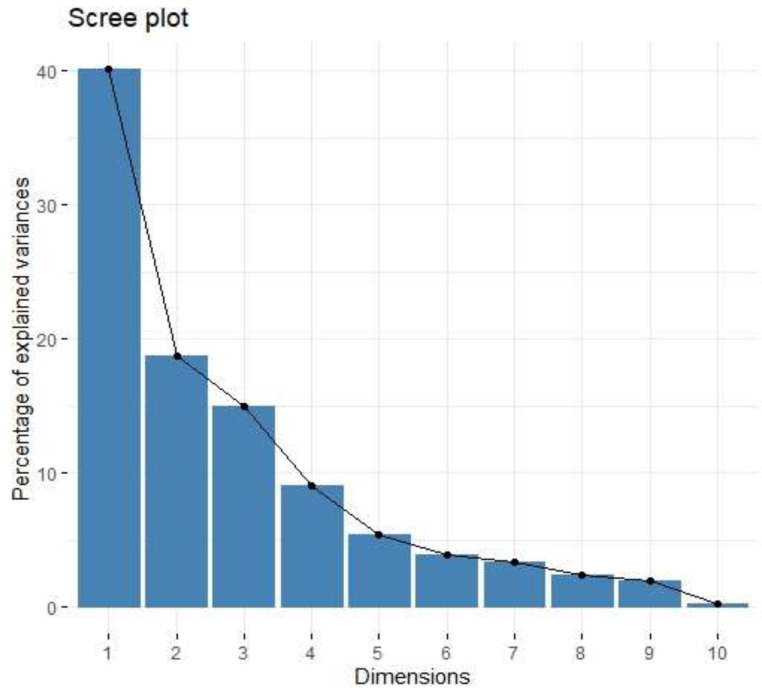
Scree plot of percentage of explained variances for 10 principal components from the PCA for protumour cytokines.

**Figure 9 cancers-15-03136-f009:**
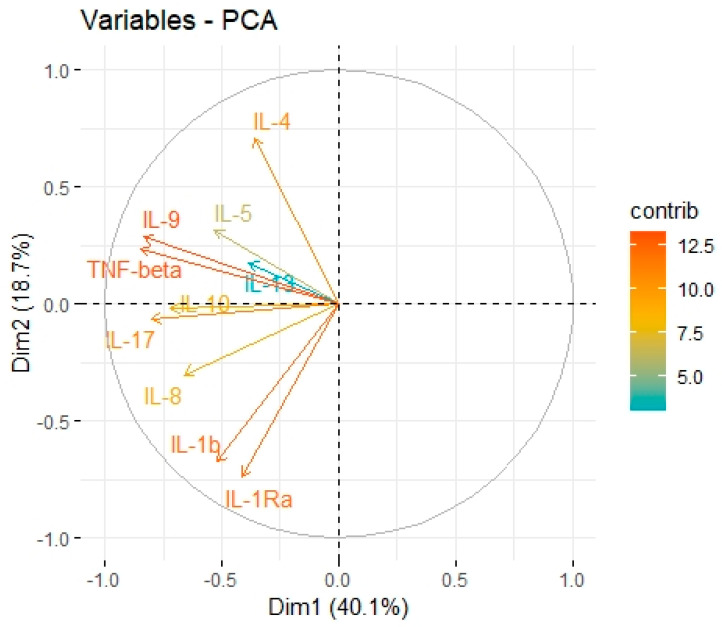
Principal components 1 and 2 represent variance in two dimensions from the PCA for protumour cytokines.

**Figure 10 cancers-15-03136-f010:**
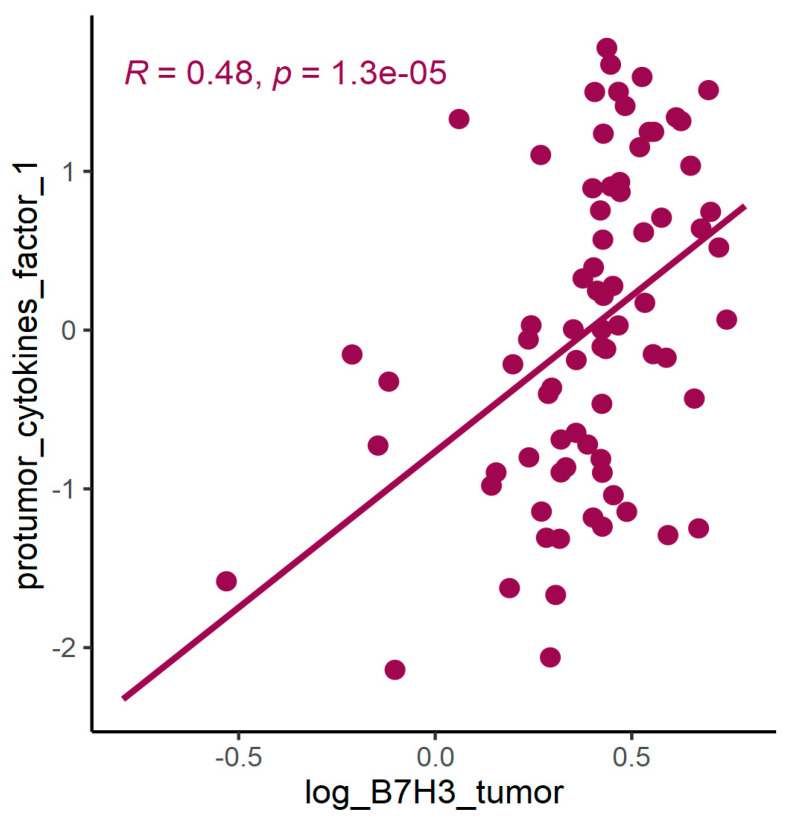
Spearman correlation between concentration of B7H3 in tumour tissue homogenates and factor 1 (principal component 1) from PCA for protumour cytokines, *p* < 0.0001.

**Figure 11 cancers-15-03136-f011:**
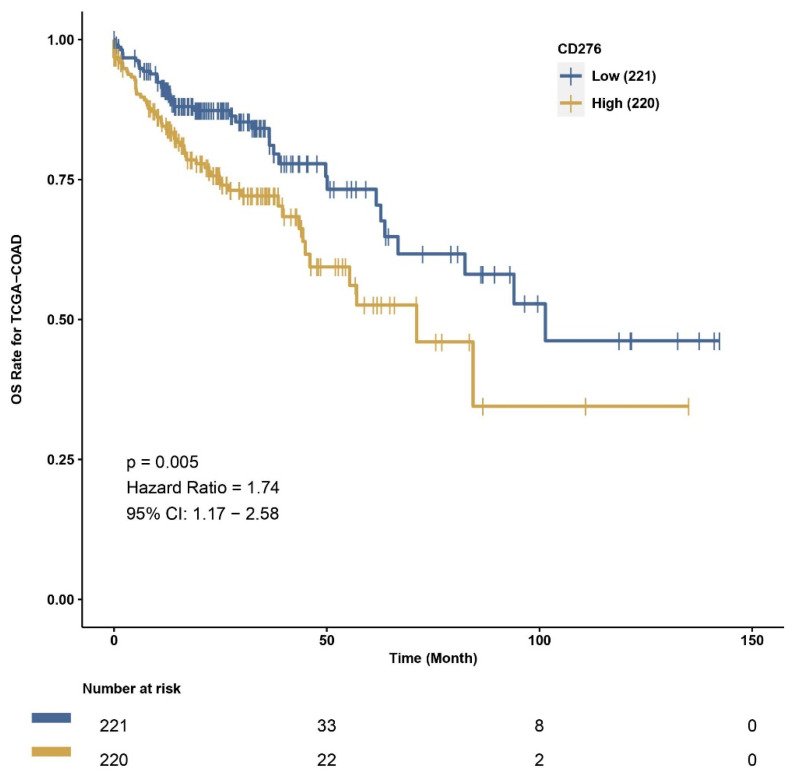
Kaplan–Meier curves demonstrating survival in high and low B7H3 expression group of CRC patients. Compared with individuals with B7H3 low expression, the B7H3 high expression group exhibited shorter survival time (*p* < 0.05)—plot obtained from CAMOIP based on TCGA-COAD dataset.

**Figure 12 cancers-15-03136-f012:**
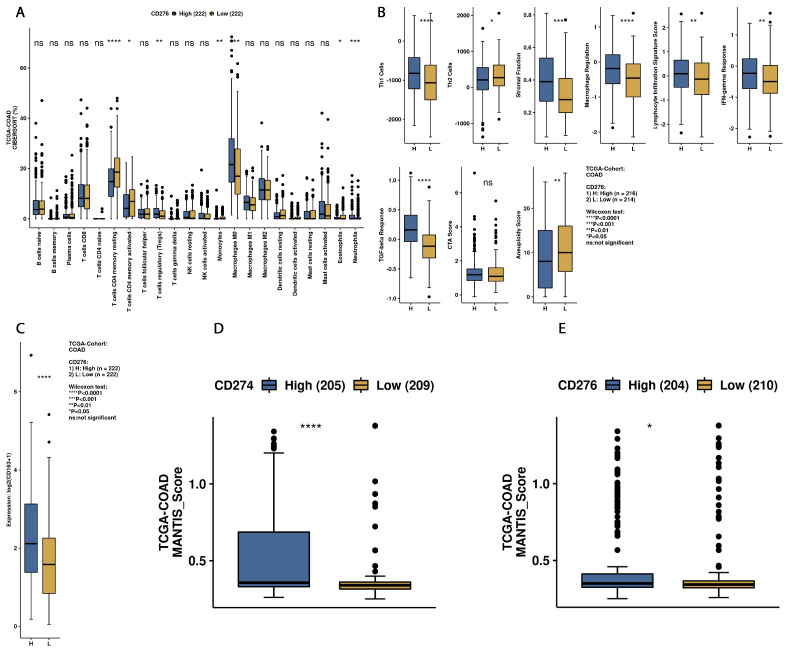
Immunological landscape in CRC tumours according to high and low B7H3 expression. analyses received from CAMOIP based on TCGA-COAD dataset. (**A**) Comparison of immune cells infiltration scores estimated by CIBERSORT algorithm between B7H3 high- and B7H3 low-expression tumours. (**B**) Immune-related scores in CRC tumours with upregulated and downregulated B7H3 expression * *p* < 0.05, ** *p* < 0.01, and *** *p* < 0.001. **** *p* < 0.0001, ns = not significant (**C**) Expression of CD163 (M2 macrophage marker) in B7H3 low and high expression groups. (**D**,**E**) Differences in MANTIS score between groups with high and low expression of B7H3 and CD274 (PD-L1). MANTIS score predicts tumour MSI status by assuming values from 0 (entirely MSS) to 2.0 (entirely MSI). The greater the MANTIS score, the more likely tumour MSI status.

**Figure 13 cancers-15-03136-f013:**
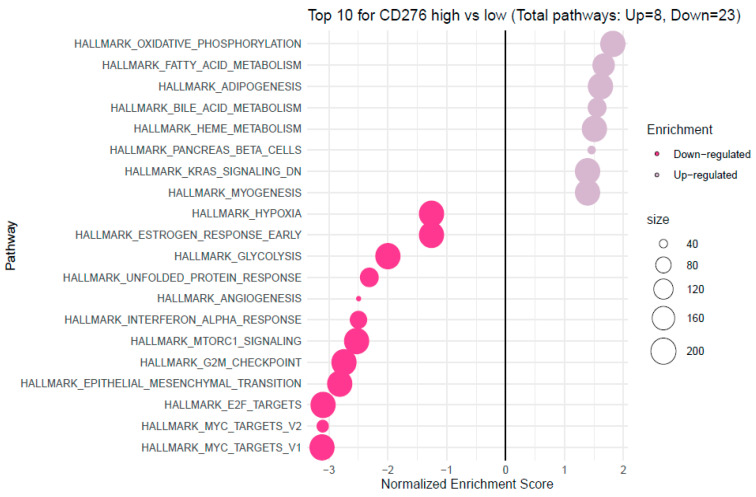
The gene set enrichment analysis (GSEA) of the association between high/low B7H3 expression in the FieldEffectCrc dataset. The hallmark pathways most impacted by high expression of B7H3 in CRC. NES: normalized enrichment score.

**Figure 14 cancers-15-03136-f014:**
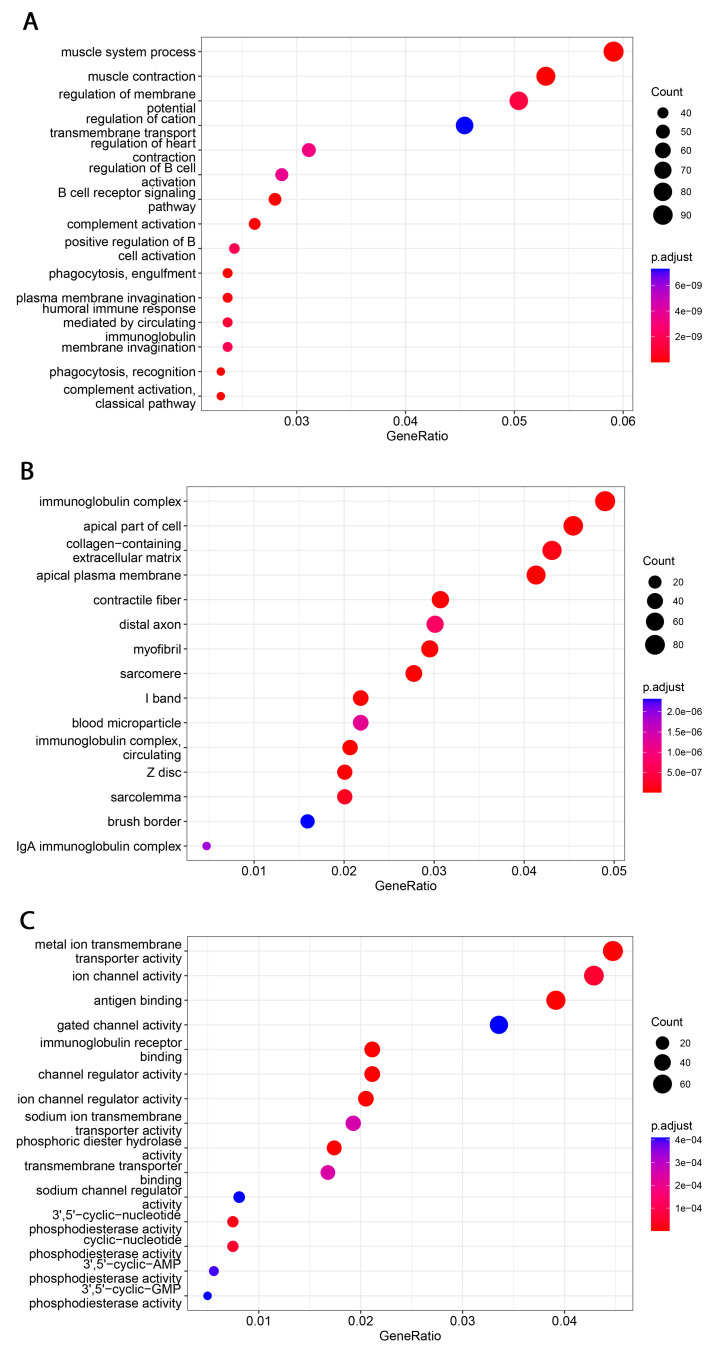
Significantly enriched GO annotations of B7H3 in FieldEffectCrc dataset. (**A**) biological processes; (**B**) cellular components; (**C**) molecular functions.

**Table 1 cancers-15-03136-t001:** Characteristics of the patients.

	Female	Male	All Cases
	72 (45.57%)	86 (54.43%)	158
Age	66.39 ± 9.32	63.83 ± 9.53	65.03 ± 9.49
tumour localization			
left-sided	48 (69.57%)	58 (71.60%)	106 (70.67%)
right-sided	21 (30.43%)	23 (28.40%)	44 (29.33%)
T parameter			
T1	1 (1.43%)	6 (7.41%)	7 (4.64%)
T2	15 (21.43%)	12 (14.81%)	27 (17.88%)
T3	44 (62.86%)	50 (61.73%)	94 (62.25%)
T4	10 (14.29%)	13 (16.05%)	23 (15.23%)
N parameter			
N0	29 (41.43%)	34 (41.98%)	63 (41.72%)
N1	28 (40.00%)	33 (40.74%)	61 (40.40%)
N2	13 (18.57%)	14 (17.28%)	27 (17.88%)
M parameter			
M0	61 (87.14%)	64 (79.01%)	125 (82.78%)
M1	9 (12.86%)	17 (20.99%)	26 (17.22%)
TNM stage			
I	12 (17.14%)	12 (14.81%)	24 (15.89%)
II	17 (24.29%)	19 (23.46%)	36 (23.84%)
III	32 (45.71%)	34 (41.98%)	66 (43.71%)
IV	9 (12.86%)	16 (19.75%)	25 (16.56%)
Grading			
low	60 (85.71%)	68 (83.95%)	128 (84.77%)
high	10 (14.29%)	13 (16.05%)	23 (15.23%)
Adjuvant treatment			
yes	7 (9.86%)	13 (15.48%)	20 (12.66%)
no	64 (90.14%)	71 (84.52%)	138 (87.34%)

**Table 2 cancers-15-03136-t002:** B7H3 IHC expression in relation to clinicopathological features of patients. Chi square test.

Characteristics	B7H3 Tumour Expression		
	Positive	Negative	*p*
All	23 (29.87%)	54 (70.13%)	-
Tumour localization			0.932
left-sided	17 (28.81%)	42 (71.19%)	
right-sided	5 (27.78%)	13 (72.22%)	
T parameter			0.327
T1	1 (25.00%)	3 (75.00%)	
T2	5 (50.00%)	5 (50.00%)	
T3	12 (23.08%)	40 (76.92%)	
T4	4 (40.00%)	6 (60.00%)	
N parameter			0.587
N0	11 (30.56%)	25 (69.44%)	
N1	9 (32.14%)	19 (67.86%)	
N2	2 (16.67%)	10 (83.33%)	
M parameter			0.156
M0	21 (31.82%)	45 (68.18%)	
M1	1 (10.00%)	9 (90.00%)	
TNM stage			0.313
I	5 (41.67%)	7 (58.33%)	
II	5 (22.73%)	17 (77.27%)	
III	11 (34.38%)	21 (65.63%)	
IV	1 (10.00%)	9 (90.00%)	
Grading			0.115
low	17 (25.76%)	49 (74.24%)	
high	5 (50.00%)	5 (50.00%)	
MSS/MSI status			0.52
MSS tumours	20 (29.85%)	47 (70.15%)	
MSI tumours	2 (20.00%)	8 (80.00%)	
TILs			0.86
0–5%	11 (30.56%)	25 (69.44%)	
6–25%	6 (28.57%)	15 (71.43%)	
26–50%	5 (31.25%	11 (68.75%)	
51–75%	0 (0.00%)	3 (100.00%)	
76–100%	0 (0.00%)	0 (0.00%)	
CD8 Lymphocytes			0.128
0–5%	5 (14.71%)	29 (85.29%)	
6–25%	10 (47.62%)	11 (52.38%)	
26–50%	4 (33.33%)	8 (66.67%)	
51–75%	2 (33.33%)	4 (66.67%)	
76–100%	1 (25.00%)	3 (75.00%)	
Budding			0.637
0–4	13 (32.5%)	27 (67.5%)	
5–9	5 (21.74%)	18 (78.26%)	
>9	3 (25.00%)	9 (75.00%)	

**Table 3 cancers-15-03136-t003:** Eigenvalue, percentage of explained variance for 3 factors (principal components) from the PCA for cytokines associated with M2 macrophages.

Factor	Eigenvalue	Variance (%)	Cumulative Variance (%)
Factor 1	4.01	44.52	44.52
Factor 2	1.05	11.68	56.20
Factor 3	1.01	11.23	67.43

**Table 4 cancers-15-03136-t004:** Loadings of 3 factors (principal components) after varimax rotation (PCA for cytokines associated with M2 macrophages).

Variable	Factor 1	Factor 2	Factor 3
IL-1a	0.64	0.28	0.11
IL-1b	0.49	0.05	0.71
IL-1ra	0.02	0.23	0.93
IL-6	0.47	0.41	0.27
IL-8	0.64	0.13	0.33
IL-10	0.06	0.93	0.14
TNF-alpha	0.44	0.74	0.15
VEGF-A	0.85	0.07	−0.02
Basic FGF	0.06	0.15	0.04

**Table 5 cancers-15-03136-t005:** Eigenvalue, percentage of explained variance for 3 factors (principal components) from the PCA for protumour cytokines.

Factor	Eigenvalue	Variance (%)	Cumulative Variance (%)
Factor 1	4.01	40.13	40.13
Factor 2	1.87	18.72	58.85
Factor 3	1.49	14.93	73.78

**Table 6 cancers-15-03136-t006:** Loadings of 3 factors (principal components) after varimax rotation (PCA for protumour cytokines).

Variable	Factor 1	Factor 2	Factor 3
IL-1b	0.05	0.92	−0.01
IL-1Ra	−0.03	0.70	0.41
IL-4	0.46	−0.23	−0.18
IL-5	0.05	0.17	0.18
IL-8	0.45	0.70	−0.14
IL-9	0.90	0.11	0.32
IL-10	0.18	0.27	0.78
IL-13	0.34	−0.23	0.78
IL-17	0.38	0.46	0.44
TNF-Beta	0.91	0.18	0.24

## Data Availability

The data can be shared up on request.

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
