# Peer review of "B7H3 Role in Reshaping Immunosuppressive Landscape in MSI and MSS Colorectal Cancer Tumours"

_cancers, 2023, doi:10.3390/cancers15123136_

Round 1

Reviewer 1 Report

The manuscript aims to explore potential marker for colorectal colon cancer and study the pathway of the new marker B7H3. However, the paper need to be improved and reorganized.

1. the image of Fig3 is in low quality, and need to enhance the resolution. Besides, the B7H3 expression in margin region should be also revealed and compared.

2. the trend line in Fig7B and Fig10 are wired and should be redraw in a statistic software.

3. the figure should be reorganized to make sure they are in logic, and the unrelated figures should be removed.

Minor language editing can be done

Author Response

Dear Reviewers,

It is our pleasure to resubmit our original article after revision entitled:

 B7H3 role in reshaping immunosuppressive landscape in MSI and MSS colorectal cancer tumours

(previous title: B7H3 expression in colorectal cancer is independent of MSI/MSS status and is related to immunosuppressive landscape in tumour microenvironment)

We have read the suggestions of the Reviewers for which we are really greatful. We considered all the issues mentioned in the reviewers’ comments. Below are the changes we have made. We want to add that our manuscript has undergone major English revision.

Response to Reviewer 1 Comments

The manuscript aims to explore a potential marker for colorectal colon cancer and study the pathway of the new marker B7H3. However, the paper need to be improved and reorganized.

Point 1: the image of Fig3 is in low quality, and need to enhance the resolution. Besides, the B7H3 expression in margin region should be also revealed and compared.

We corrected the resolution of Fig3 and added a picture showing negative staining for B7H3 in tumour tissue. Unfortunately, we cannot provide a figure presenting B7H3 staining in tumour free margin tissue because B7H3 expression was assessed with the use of IHC method only in tumour specimens which were thick sections of a representative formalin fixed, paraffin-embedded (FFPE) tumour tissue block. To compare B7H3 expression in tumour and margin, we measured concentration of B7H3 protein in tissue homogenates using commercially available ELISA test (quantitative method), so in the manuscript we showed the difference in B7HH3 protein concentration between tumour and margin (Figure 1).

Figure 3. B7H3 immunostaining in CRC specimens. (A) B7H3 positive staining (B, C) B7H3 negative staining. (C) Tumour infiltrating lymphocytes (TILS) are highlighted  in frame (Opta Tech 2200 Camera, magnification 200×)

Point 2: the trend line in Fig7B and Fig10 are wired and should be redraw in a statistic software.

All Figures in the manuscript were corrected and now are provided in higher resolution.

Point 3: The figure should be reorganized to make sure they are in logic and unrelated figures should be removed.

Results section was redrafted to provide more clear arrangement of figures and text. We hope that the results in present form are better organized.

Reviewer 2 Report

This manuscript is very well written. I recommend authors to make figure labels more clear. At present, they are very blur and hard to read. 

Author Response

Dear Reviewers,

It is our pleasure to resubmit our original article after revision entitled:

 B7H3 role in reshaping immunosuppressive landscape in MSI and MSS colorectal cancer tumours

(previous title: B7H3 expression in colorectal cancer is independent of MSI/MSS status and is related to immunosuppressive landscape in tumour microenvironment)

We have read the suggestions of the Reviewers for which we are really grateful. We considered all the issues mentioned in the reviewers’ comments. Below are the changes we have made. We want to add that our manuscript has undergone major English revision.

Response to Reviewer 2 Comments

Point 1: This manuscript is very well written. I recommend authors to make figure labels more clear. At present, they are very blur and hard to read.

All Figures in the manuscript were redrawn in statistic software  and are provided in higher resolution.

Reviewer 3 Report

This study assessed the expression of B7H3 in relation to clinicopathological parameters, including MSI/MSS status, CD8+ T cells, histopathological features: budding,  tumor-infiltrating lymphocytes, TNM scale, and grading. Overall, the study is well-desinged and provides useful information. Therefore, I just have two minor suggestion.

1. The title should be revised and shortened.

2. English editing is requiring to improve the manuscript.

3. Revised the following paragraph (Line 493-6)
Authors should discuss the results and how they can be interpreted from the perspective of previous studies and of the working hypotheses. The findings and their implications should be discussed in the broadest context possible. Future research directions may also be highlighted

required minor editing

Author Response

Dear Reviewers,

It is our pleasure to resubmit our original article after revision entitled:

 B7H3 role in reshaping immunosuppressive landscape in MSI and MSS colorectal cancer tumours

(previous title: B7H3 expression in colorectal cancer is independent of MSI/MSS status and is related to immunosuppressive landscape in tumour microenvironment)

We have read the suggestions of the Reviewers for which we are really grateful. We considered all the issues mentioned in the reviewers’ comments. Below are the changes we have made. We want to add that our manuscript has undergone major English revision.

Response to Reviewer 3 Comments

This study assessed the expression of B7H3 in relation to clinicopathological parameters, including MSI/MSS status, CD8+ T cells, histopathological features: budding,  tumour-infiltrating lymphocytes, TNM scale, and grading. Overall, the study is well-desinged and provides useful information. Therefore, I just have two minor suggestion.

Point 1. The title should be revised and shortened

The manuscript title were revised and was changed to  “ B7H3 role in reshaping immunosuppressive landscape in MSI and MSS colorectal cancer tumours”. We hope that in the present form the title of article is more appropriate.

Point 2. English editing is requiring to improve the manuscript.

We submitted the manuscript for language editing.

Point 3. Revised the following paragraph (Line 493-6)

Authors should discuss the results and how they can be interpreted from the perspective of previous studies and of the working hypotheses. The findings and their implications should be discussed in the broadest context possible. Future research directions may also be highlighted

The paragraph were revised and after changes in the manuscript is provided in Line 517-33):

Our findings improved understanding of B7H3 role in cancer immunity. In addition, the studies on colorectal tumour tissues allowed us to estimate clinical, pathological, and immunological parameters. We assessed B7H3 concentrations  with ELISA, and IHC, determined MSI/MSS status, and estimated the immune landscape by cytokine screening panel and tumour-infiltrating CD8+ T cells. The part of analysis based on mRNA expression profiles has significantly expanded the scope of research, reporting many significant molecular roles of B7H3. CAMOIP, a tool for comprehensive analysis of potential immunotherapy targets, allowed us to perform survival analysis according to low or high B7H3 expression in the TCGA cohort and determined the link between B7H3 expression, immune cells infiltration, Mantis score, and patients survival in CRC tumours. , We studied the pathways activated in tumours with upregulated B7H3 expression with Gene Set Enrichment Analysis. Our future research direction focuses on further studies  of B7H3 as a potential target for immunotherapy, which requires extending the research to cellular, animal, and clinical models. The other cancer therapy trends require studying the diagnostic and therapeutic potential of B7-H3 thoroughly, its cellular and molecular mechanisms, B7H3-mediated metastases, tumour-associated vasculature, and recurrence. Clinical data with a long-term follow-up are essential to understand the relevance of B7H3 expression.

Reviewer 4 Report

The research article under the title “B7H3 expression in colorectal cancer is independent of MSI/MSS status and is related to immunosuppressive landscape in tumor microenvironment” by Mielcarska and coworkers presents interesting results. The results are presented in a clear way with statistical analysis done well. My recommendation is MAJOR REVISION and the article will be suitable for publication after these questions are addressed by the authors:

1.      Line 66 – the authors should give more details on the B7 family of compounds from a structural point of view, just to explain how this structure is related to activity

2.      Line 102 – there should be a space between a number and unit, this authors should check this throughout the manuscript

3.      The first part of results should give a brief overview of the selected patients groups and their characteristics

4.      Figure 2 shows data that are quite spread through the graph, the authors should mention some of the limitations, as it is not quite clear if the dependency increases or decreases

5.      The authors should confirm main findings with the other similar methods

Round 2

Reviewer 1 Report

The authors have addressed all my concerns, and I would like to endorse the paper to be published.

Reviewer 4 Report

the authors have answered all of the queries by the Reviewer. The article is suitable for publication.